# Restoration of T and B Cell Differentiation after RAG1 Gene Transfer in Human RAG1 Defective Hematopoietic Stem Cells

**DOI:** 10.3390/biomedicines12071495

**Published:** 2024-07-05

**Authors:** Nataël Sorel, Francisco Díaz-Pascual, Boris Bessot, Hanem Sadek, Chloé Mollet, Myriam Chouteau, Marco Zahn, Irene Gil-Farina, Parisa Tajer, Marja van Eggermond, Dagmar Berghuis, Arjan C. Lankester, Isabelle André, Richard Gabriel, Marina Cavazzana, Kasrin Pike-Overzet, Frank J. T. Staal, Chantal Lagresle-Peyrou

**Affiliations:** 1Human Lymphohematopoiesis Laboratory, Université Paris Cité, Imagine Institute, INSERM UMR 1163, 75015 Paris, Franceisabelle.andre@inserm.fr (I.A.); 2ProtaGene CGT GmbH, Im Neuenheimer Feld 582, 69120 Heidelberg, Germany; 3Biotherapy Clinical Investigation Center, Groupe Hospitalier Universitaire Ouest, AP-HP, INSERM, 75015 Paris, France; 4Department of Immunohematology and Blood Transfusion, L3-Q Leiden University Medical Center, 2333 ZA Leiden, The Netherlands; 5Department of Pediatrics, Leiden University Medical Center, 2333 ZA Leiden, The Netherlands; d.berghuis@lumc.nl (D.B.); a.lankester@lumc.nl (A.C.L.); 6Biotherapy Department, Necker-Enfants Malades Hospital, AP-HP, 75015 Paris, France; k.pike-overzet@lumc.nl; 7Imagine Institute UMR1163, Université Paris Cité, Sorbonne Paris Cité, 75015 Paris, France

**Keywords:** gene therapy (GT), recombination activating gene 1 (RAG1), severe combined immunodeficiency (SCID)

## Abstract

Recombinase-activating gene (RAG)-deficient SCID patients lack B and T lymphocytes due to the inability to rearrange immunoglobulin and T cell receptor genes. The two *RAG* genes act as a required dimer to initiate gene recombination. Gene therapy is a valid treatment alternative for RAG-SCID patients who lack a suitable bone marrow donor, but developing such therapy for RAG1/2 has proven challenging. Using a clinically approved lentiviral vector with a codon-optimized *RAG1* gene, we report here preclinical studies using CD34+ cells from four RAG1-SCID patients. We used in vitro T cell developmental assays and in vivo assays in xenografted NSG mice. The RAG1-SCID patient CD34^+^ cells transduced with the RAG1 vector and transplanted into NSG mice led to restored human B and T cell development. Together with favorable safety data on integration sites, these results substantiate an ongoing phase I/II clinical trial for RAG1-SCID.

## 1. Introduction

The recombination activating gene 1 (*RAG1*) encodes for the RAG1 protein and it is a key master gene controlling the rearrangement of T and B cell receptors essential for T and B lymphocyte development and function [1]. More recently, RAG1 has been reported as a regulator of immune tolerance [2,3]. Thanks to advances in genome sequencing technology for primary immunodeficiency diagnosis [4], it is now clearly admitted that RAG1 mutations are associated with a broad spectrum of clinical and immunological manifestations ranging from combined immune deficiency to severe combined immunodeficiency (SCID) [5,6]. The RAG1-SCID groups represent up to 14% of all the SCID [7] and this rare autosomal recessive form is characterized by a total absence of T and B cells associated with infections and failure to thrive early in life. Until recently the only curative treatment for this life-threatening disorder was allogenic bone marrow hematopoietic stem cell transplantation (HSCT). This strategy restores functional T and B lymphoid compartments with a survival rate of over 70% in the presence of an HLA-match donor [8]. Unfortunately, the majority of RAG1-SCID has mismatched or unrelated donors and in this context, 20% of the RAG1 treated patients have severe complications like a poor chimerism or a graft versus host disease. These adverse effects could compromise the outcome of HSCT, the immune reconstitution and patient survival [9,10,11].

We made the assumption that ex vivo *RAG1* gene transfer into bone marrow hematopoietic stem and progenitor cells (HSPCs, referred to hereafter as CD34+) from RAG1-deficient patients represents an alternative therapeutical approach that could overcome the drawbacks of conventional HSCT. Indeed, autologous HSC-base gene therapy (GT) was developed more than 30 years ago and was effective for several primary immunodeficiencies [12,13,14].

Different attempts in mouse models underlined that gamma (γ)-retroviral vectors and native *RAG1* transgene are not efficient enough to bring full immune reconstitution. Moreover, with these constructs, the integration of a very high copy number (VCN > 5) that could increase the risk of insertional mutagenesis has not been adapted in future clinical applications [15,16,17]. The use of defective lentiviral vectors (SIN-LV) and a codon-optimized *RAG1* transgene (co RAG1) improved the gene transfer strategy and pointed out that a high *RAG1* transgene expression is required for a functional T and B lymphocyte compartment [18,19]. To confirm the transduction efficiency, a new SIN-LV MND-co RAG1 vector was introduced into RAG1-deficient HSPCs. In the RAG1 mouse model and in the xenogeneic NSG model, *RAG1* transgene expression restored functional T and B cell compartments with a VCN compatible with clinical use (VCN± 1) and no significant toxicity was observed [20]. Therefore, the MND-co RAG1 construct was considered as the best choice for this gene therapy approach and a RAG1 pre-GMP lentiviral batch was produced [20] for the preclinical studies.

With the preclinical studies herein described, we demonstrated the safety of the RAG1 pre-GMP lentiviral vector associated with an efficient ability to transduce human CD34+ and to restore stable lymphocyte differentiation and function both in vitro and in vivo. Necessitated by the rarity of this disease, we could perform the experiments with only five RAG1 patient samples; nevertheless, these studies reported here also allowed for a detailed analysis of T cell development in the thymus. These encouraging data constituted an essential step to launch the first RAG1 gene therapy clinical trial (phase I/II) in the Netherlands, which recruits patients from all of Europe, Turkey and Australia and paves the way for an alternative curative treatment dedicated to RAG1-SCID patients without any compatible donors. Importantly, the phase I/II clinical trial is now open for patients worldwide with RAG1 mutation but lacking an HLA-matched donor.

## 2. Materials and Methods

### 2.1. Human Samples

The study was conducted in accordance with the French and NL legislation and the principles of the Helsinki declaration. Human umbilical CB was obtained from the Biological Resources Center of Saint Louis Hospital (Paris, France) in accordance with the ethically approved procedures (convention 23 September 2014). The bone marrow (BM) samples from healthy donors (HD) were the unused fraction of grafts when the HD had provided their informed consent for research use. For RAG1-deficient patients, informed consent was obtained from the parents or legal wards of the patients and the study protocol was approved by the regional investigational review board ethical committee and Ministry of Research (DC-2020-3994). The Dutch patient is part of an ongoing CT which has received approval from the Dutch Central Committee for studies in humans (CCMO, NL70818.000.19). Mononuclear cells were isolated by density separation on Lymphoprep. CD34^+^ HSPCs were magnetically enriched (purity > 93%) from CB and BM samples, as described previously [21].

### 2.2. Transduction and Cell Culture

After an overnight pre-activation in a cocktail of human cytokines (300 ng/mL stem cell factor (SCF), 300 ng/mL FMS-like tyrosine kinase 3 ligand (Flt3-L), 100 ng/mL Thrombopoietin (TPO) and 10 ng/mL Interleukin-3 (IL-3), all from PeproTech, Inc., Rocky Hill, NJ, USA), CD34+ cells were transduced with the RAG1-GMP lentiviral batch according to the protocol described in [20]. After the transduction step, cells were either injected into NSG mice or cultured with the cytokines cocktail for 16 days or in a T cell differentiation system for up to 35 days. For T cell differentiation, CD34+ cells were first cultured on a DL4-coated plate for 7 days and then transferred on OP9-DL1 culture as previously described [22].

### 2.3. Vector Copy Number (VCN) and RAG1 Transgene Expression

DNA and RNA extraction was performed according to the protocols provided in the Quick DNA/RNA Microprep plus kit (Zymo research, Ref: D7005, Tustin, CA, USA), the DNeasy Blood and Tissue kit (Qiagen, Ref: 69506, Hilden, Germany) or the RNeasy micro kit (Qiagen, Ref: 74004) kit. The integrated vector copy number (VCN) in genomic DNA was assessed by droplet digital polymerase chain reaction (ddPCR) with Bio-rad QX200 ddPCR System (Hercules, CA, USA). Droplets were generated with the generator and PCR was performed using TaqMan PCR Master Mix (Thermo Fisher Scientific, St. Louis, MO, USA). Fluorescence detected in PCR-positive droplets was quantified according to the Poisson distribution, and VCN was calculated according to (PSI × 2)/(ALB).

The transgene expression (coRAG1) was determined on cDNA (High capacity cDNART kit, Applied Biosystems, Ref: 4368814, Waltham, MA, USA) using TaqMan gene expression master mix (Applied Biosystems, Ref: 4369016) and *RAG1* specific probes. CoRAG1 expression was normalized with a GAPDH probe. The data were analyzed using the comparative cycle threshold method. It is presented as the relative quantification (RQ) of the coRAG1 expression normalized against the calibrator sample corresponding to the non-transduced condition (RQ = 2^−ΔΔCt^).

### 2.4. Human Primer and Probe Sequences

#### 2.4.1. Sequence for the Droplet Digital PCR

Albumin: probe VIC sequence: CCTGTCATGCCCACACAAATCTCTCC/forward primer: GCTGTCATCTCTTGTGGGCTGT/reverse primer: ACTCATGGGAGCTGCTGGTTC

PSI: probe FAM sequence: CGCACGGCAAGAGGCGAGG/forward primer: TCCCCCGCTTAATACTGACG/reverse primer: CAGGACTCGGCTTGCTGAAG

#### 2.4.2. Sequence for RT-Quantitative PCR

GAPDH: Hs00266705_g1 Taqman Gene Expression Assays, Applied Biosystems, Ref: 4331348

Endogenous RAG1: probe FAM-NFQ-MGB sequence: CCATTGTGCCTTATGCT/forward primer: GCCAAACCTAACTCTGAACTGTGT/reverse primer: GCGTCTCGTGGTCAGACTCA

CoRAG1: probe FAM-NFQ-MGB sequence: AGAGTGTGCATCCTGCGGTGCCT/forward primer: CAACTGCAAGCACGTGTTCT/reverse primer: GCAGTAGCTGCCCATCACTTT

### 2.5. Adoptive Transfer into NSG Mice

NOD-SCID-γc-/- strain (NSG) mice were obtained from Charles River companies and maintained in a germ-free facility (agreement 75-15-35). An average of 2 to 6 × 10^5^ RAG1-transduced or not transduced CD34+ cells were injected in adult NSG mice (6 to 8 weeks old) previously conditioned with 45–50 mg/kg of a total busulfan dose. Engraftment in bone marrow (BM), spleen and thymus were analyzed after 16 weeks by flow cytometry. DNA and RNA were also extracted from the organs for further analysis.

In total, 1 × 10^6^ CD7^+^ cells generated from 7-day DL-4 cultures were intrahepatically transplanted into 4-day-old NSG mice. Six weeks post-transplantation, the thymi were analyzed by flow cytometry for human cell engraftment and thymopoiesis. At this time point, human cell engraftment was undetectable in the other hematopoietic organs.

All the animal procedures were approved by the animal committee of the University of Paris (Paris, France; 16 February 2021) and the French Ministry of Research (APAFIS#29592-2020120216106476). The procedures were performed in accordance with European Union (EU) Directive 2010/63/EU. All animal experiments using patient P4 CD34+ cells were approved by the Dutch Central Commission for Animal Experimentation (Centrale Commissie Dierproeven, CCD, 2022).

### 2.6. Flow Cytometry Analysis

For the staining, cells were incubated with the appropriate antibodies (detailed in Appendix A) for 15 min in ice, washed and then resuspended in FACS buffer. All flow cytometry data were acquired with a Gallios flow cytometer (Beckman Coulter, Krefeld, Germany) and then analyzed using FlowJo software (version 10.2, Treestar, Ashland, OR, USA) or Kaluza software (version 2.1, Beckman Coulter, Brea, CA, USA).

During the FACS analysis, all the gatings were conducted on live cells, determined by the exclusion of 7-AAD-positive cells. CD34^+^CD19+ and CD34^+^CD19-BM cells were sorted using a FACS Aria II SORP cell sorter (BD Biosciences, Franklin Lakes, NJ, USA).

### 2.7. Serum Analysis

The presence of human IgM was evaluated in the serum of NSG mice. IgM level was evaluated by a sandwich-enzyme-linked immunosorbent assay (ELISA) using a quantification ELISA kit (Bethyl Laboratories, Montgomery, TX, USA) according to the manufacturer’s instructions. Data was acquired at a wavelength of 415 nm using a Bio-Rad microplate. Antibody concentration was calculated by using human reference serum as standard (Bethyl Laboratories, Montgomery, TX, USA).

### 2.8. Analysis of Vector Integration Sites

DNA was extracted and S-EPTS/LM-PCR [23] was performed on each sample using up to 500 ng of DNA per replicate. Sequencing was performed using MiSeq technology (Illumina). Sequencing data were analyzed using GENE-IS [24], which allows the detection of IS mappable to unique locations in the genome.

Cancer-associated genes were obtained from the Catalogue of Somatic Mutations in Cancer (COSMIC) Cancer Gene Census. This list includes the top 2 categories (tier 1 and tier 2) of cancer-associated genes (http://cancer.sanger.ac.uk/census, accessed on 1 September 2019; v90) [25]. Cancer gene annotations (chromosome, gene start, gene end, strand, gene name, transcript count, TSS and transcript support level (TSL)) were obtained from Ensembl (http://www.ensembl.org/biomart/martview, accessed on 20 March 2020; dataset human genes, version GRCh38.p13).

A systems biology approach was used to dissect biologically relevant IS clusters, termed common integration sites (CISs) [26]. CIS analysis allows the identification of positional IS accumulations that are statistically unlikely to occur by chance [27,28]. Any IS found in a sample was associated with a node that contained the location of the IS. If the distance of two nodes was less than a threshold distance (50 kb) the two nodes were connected. The arbitrary 50 kb threshold was selected using the influence window size, where a causal relation is found between an integration event and gene expression [29]. Each CIS is composed of a defined number of ISs and has a defined dimension between the two outmost positioned ISs.

### 2.9. TCR Repertoire Sequencing and Analysis

To analyze the TCR repertoire of samples, RACE-PCR was performed [30]. Briefly, up to 500 ng of RNA was used as input for cDNA synthesis using a TCR constant region-specific primer followed by the addition of an adaptor oligo molecule at the 5′cDNA end. cDNA was purified using AMPure XP and amplified by two subsequent nested PCR reactions using primers binding to the respective chain constant region and the introduced adaptor. In the second amplification step, fusion primers enabling sequencing on Illumina’s MiSeq platform were used and PCR products were purified using AMPure XP beads. Finally, products were pooled and sequenced using MiSeq technology (Illumina). Sequencing data were analyzed using MiGEC, MiXCR [31] and VDJtools [32].

### 2.10. Statistical Analysis

The results were analyzed in one-way repeated-measures (RM) analysis of variance (ANOVA) or using the Mann–Whitney rank-sum test. The data were plotted as the mean ± SEM, using GraphPad Prism software (version 5, San Diego, CA, USA). The threshold for statistical significance was set to *p* < 0.05.

## 3. Results

### 3.1. Transduction of HD CD34+ with the RAG1 Lentiviral Batch

CD34+ isolated from cord blood (CB) or healthy donor bone marrow (HD) were transduced (T) or not (NT) with the RAG1-lentiviral pre-GMP vector (as described in Garcia-Perez L et al., 2020 [20]). We first tested the impact of the vector concentration (9.5, 16, 30 and 40·10^8^ infectious genomes (Ig)/mL) while keeping a constant multiplicity of infection (MOI) in all the conditions. After a 14-day culture in the presence of cytokines, the cell proliferation and the *RAG1* transgene integration, assessed by the mean vector copy number (VCN), were evaluated. In all conditions, the vector concentration had no impact on the cell’s proliferative index (Appendix A) and on the transgene integration suggesting that the virus-diluting buffer is not directly linked to the integration efficacy. Of note, the mean VCN is slightly higher in the CB cells than in HD (Appendix A), but in both conditions, we observed a good correlation between the VCN and *RAG1* transgene expression (Appendix A). To test the T cell differentiation capacity of transduced CD34+ overexpressing RAG1, cells were seeded on the DL4 and OP9-DL1 systems for 7 and 28 days, respectively. By flow cytometry, we evaluated the number of early T cell progenitors expressing CD7+ marker and more mature T cells co-expressing CD4+CD8+ markers (Appendix A). In the NT and T conditions, CD7+ pro-T cells or CD4+CD8+ cells are found in the same proportion. Lastly, the percentage of mature CD3+TCRgd+ cells did not differ between all the conditions (Appendix A). Altogether, these data demonstrate that *RAG1* transgene overexpression has no impact on the in vitro T cell differentiation of CD34+ from CB or bone marrow origin.

To confirm the efficacy of the gene transfer strategy CD34+, NT or T bone marrow CD34+ from HD were injected into NOD/SCID/γc^-/-^ (NSG) mice, previously conditioned with busulfan. Human chimerism, T and B lymphocyte reconstitution were evaluated by flow cytometry 4 months after transplantation. Human chimerism and human B cells were observed in the bone marrow (BM) and spleen of all mice (Appendix A) without any significant difference between the two groups (NT or T). Of note, human thymus engraftment and double positive CD4+CD8+ cells (DP) number were more difficult to obtain and found only in half of the transplanted mice (Appendix A). However, the presence of CD4+CD8+ confirms that *RAG1* overexpression in HD CD34+ does not prevent their capacity to mature along the T cell lineage. Altogether, these data demonstrate that the RAG1-lentiviral pre-GMP batch efficiently transduces human CD34+ without any toxicity and the VCN obtained both in vitro and in vivo is compatible with a gene therapy clinical approach for RAG1-deficient patients.

### 3.2. Transduction and Follow-Up of RAG1 Deficient Bone Marrow Sample In Vitro

Several bone marrow CD34+ cells were obtained from pediatric patients with RAG1 mutations (Table 1). The phenotype of the bone marrow samples from P1, P2 and P3 was analyzed by flow cytometry. In all samples, the CD34+ population contained a high proportion of pro-B cells (CD34+CD19+) compared to the healthy donor (HD) counterpart (Figure 1a), an observation in accordance with the B cell differentiation blockage already described for the RAG1-SCID form. CD34+ from HD or RAG1 patients were transduced (T) or not (NT) with the RAG1-lentiviral GMP vector batch. The day following the transduction step, cell expansion was identical in the NT and T conditions but lower in the patient’s cells when compared to the HD controls. This observation was correlated with the drastically reduced percentage of the CD34+CD19+ subset in all the patient samples (Figure 1b).

The purified CD34^+^ from P1 and P2 samples were cultured in vitro for 12 days within a cytokine cocktail (CK) or on DL4 and DL1 systems for 7 days and up to 35 days, respectively. The proliferative capacity, measured in three independent experiments for P1 (referred as exp1 and exp2) and P2 patients, did not reveal major differences between the NT and T conditions. Of note, in both culture conditions (cytokine or DL4), P2 cells’ expansion was reduced as compared to P1 (Figure 2a,b). After the 7-day culture on the DL4 system, most of the CD34+ differentiated toward the pro-T cells subset expressing the CD7 marker. In each individual experiment, the percentage and the number of CD7+ pro-T cells were nearly identical in the NT and T conditions (Figure 2c). The pro-T cells were then cultured on the OP9-DL1 system to evaluate late T cell differentiation. In a time-course kinetic performed for P1 samples, we observed a higher percentage and number of CD4+CD8+ differentiated cells (up to 44%) in the T condition as compared to the NT counterpart (Figure 2d). For the P2 sample, as observed in the cytokine cocktail culture, cell expansion was limited during both the DL4 and DL1 cultures and it was not possible to quantify the cell number at D14 and D21. Nevertheless, CD4+CD8+ expressing cells were observed with a percentage of DP higher and stable over time in the T condition (Figure 2e).

In all the cultures mentioned herein, we measured *RAG1* transgene integration with a droplet digital-PCR method. In the two experiments performed with P1 samples, an identical VCN was found in the cytokine cocktail culture (1.96 for exp1 and 1.8 for exp2) with a two-fold decrease after the T cell differentiation culture (0.4 for exp1 and 0.8 for exp2). Conversely, for the P2 sample, a higher VCN was obtained but the differential in terms of VCN variation between the CK (5) and DL4 (3.9) cultures is similar to the ones found for P1. *RAG1* mRNA transgene, measured by a quantitative RT-PCR method, was highly expressed (Figure 3) in all the transduced conditions as compared to the NT one. Altogether, these results highlight that *RAG1* transgene expression is detected in all transduced CD34+ with a VCN compatible with the restoration of T cell differentiation.

### 3.3. In Vivo Follow-Up of RAG1 Transduced Cells from RAG1 Deficient Patients

For the in vivo experiments, P1 and P2 RAG1-transduced CD34+ cells were injected into adult NSG mice that received a busulfan conditioning regimen. Human chimerism and immune reconstitution were evaluated by flow cytometry 16 weeks after transplantation. Due to the low number of P1 bone marrow cells available after the transduction step, only two NSG mice (P1-NT and P1-T) issued from the first experiment (exp1) were injected. Human chimerism was found relatively similar in both NT and T mice and higher in the BM as compared to the spleen and thymus. The number of immature (CD19^+^IgM^+^) medullar B cells was 2.5 fold higher in the T condition as compared to the NT condition (Table 2) and associated with a higher percentage of CD90+ cells (Figure 4a) suggesting that human stem cell progenitors with self-renewal capacities are present. In the spleen, the mature (CD19^+^IgM^+^IgD^+^) B cell compartment was detectable only in the NSG mouse injected with RAG1-transduced cells (Figure 4b). These cells were functional as assessed by the presence of human IgM in the serum (Table 2). In the thymus, human cell number was low in the T condition but all the maturation stages (CD4+CD8+, CD4+SP, CD8+SP) were represented (Figure 4c). The VCN of 0.46 and 2.5 detected in the BM and spleen, respectively, validated the presence of human cells expressing the *RAG1* transgene (Figure 4d). These cells are functional as the human B and T compartments were restored confirming the efficacy of the *RAG1* gene transfer strategy in CD34+ cells (Figure 4d).

To go further into the T cell functional analysis, the TCR beta (TCRB) and alpha (TCRA) repertoire diversity were analyzed in the spleen samples even if mature T cells were nearly undetectable by flow cytometry (<2%). In the NT condition, 11 and 22 TCRB clones were detected for replicate 1 and 2, respectively, whereas in the RAG1-transduced condition, 223 (replicate 1) and 73 (replicate 2) were detected. Of those, the most abundant clone had a frequency of 6.6%. Similar results were obtained for the TCRA repertoire, where 5 and 18 clones were detected in the NT conditions and in the RAG1-transduced condition with 77 and 90 clones for replicate 1 and 2, respectively (Table 3, Figure 4e and Appendix A). This shows that a higher TCR diversity was restored without any prominent clones (Figure 4e and Appendix A, Table 3). The CDR3 length distributions show that the T condition is more polyclonal than the NT condition. This is shown by the broader distribution and by the increased frequency of the “other” category (grey bars in Figure 4f and Appendix A), which display clones that are not among the 10 most frequent ones. Altogether, these data demonstrated that dominant clonotype frequency was reduced after RAG1 gene therapy and it is associated with an increased CDR3 length diversity.

NT and RAG1-transduced CD34+ cells from P2 were injected into busulfan-pre-conditioned NSG mice. Sixteen weeks after the injection, the mice were killed and the spleen, thymus and bone marrow were analyzed. Human chimerism was heterogeneous in the bone marrow of the 3 P2-T mice (P2-T4, P2-T6, P2-T7) but in correlation with the B cell count. Despite a low chimerism in the spleen (<3%), the mature B cell number was higher in the RAG1-transduced condition as compared to the NT counterpart, a result in accordance with the human IgM level detected in the serum (Table 2). In the thymus, human chimerism was found in three out of four injected mice and CD4+CD8+ thymocytes were detected in one out of the three mice that received RAG1-transduced cells (Figure 5a). The repertoire diversity of the TCRB and TCRA rearrangements could not be analyzed due to the low sequence numbers available. In the bone marrow and spleen of each mouse, the VCN was comparable ranging from 0.3 in mouse T6 to 1.2 in mouse T7. RAG1 transgene expression was detected in the T conditions of all samples and in both tissues, whereas it was absent in the NT conditions (Figure 5b). Human CD45+ cells were magnetically sorted from the mouse bone marrow (T4, T6 and T7) and seeded on a 7-day methylcellulose assay. The colony forming unit (CFU) did not differ between the conditions and the VCN evaluated in the CFU corroborated the one observed in the bulk BM demonstrating the stability of the *RAG1* transgene integration.

We had a unique opportunity to assess the T and B cell development after RAG1 gene transfer into P4 patient cells (P4) because of the availability of surplus cells from the clinical RAG1 trial. Cells that were not used for transplantation into the patient were used as leftover cells to assess T cell development after transplantation into busulfan-conditioned NSG mice. Briefly, mobilized CD34+ RAG1-cells were either cultured o/n with cytokines (NT) or cultured and transduced (T) with the clinical grade lentiviral batch (LV RAG1). We transplanted 150 k CD34+ cells per mouse and used five mice per condition. Non treated CD34+ cells were also used as a control (untreated). While mice showed detectable human chimerism (Figure 6a–c), the culturing of CD34+ cells diminished the percentage of chimerism by almost 3 fold in the spleen (Figure 6c). Of note, although five mice per group received cells from the same pooled condition, there was remarkable heterogeneity in all mice. Human T cell development in the thymus was variable (Figure 6b) but led to the development of mature T cells in the periphery. In the PBMCs, on average, 40% of the human cells were composed of T cells after transduction while no human T cells were detected in mice transplanted with non-transduced cells (Figure 6d). Importantly, transduction with the LV RAG1 vector was also associated with the detection of immature IgD+ cells while non-transduced CD34+ patient cells did not lead to IgD+ offspring (Figure 6c). Therefore, although variable, the in vivo reconstitution experiments showed robust B and T cell differentiation.

Altogether, the in vivo data confirm that RAG1 corrected CD34+ could differentiate into T and functional B lymphocytes without any measurable toxicity. These data support the notion that the RAG1 lentiviral pre-GMP batch is efficient in correcting RAG1-deficient CD34+ with a VCN and *RAG1* transgene expression stable over time. Accordingly, the current gene transfer protocol used in this study is compatible with the development of a clinical gene therapy trial for RAG1-SCID patients.

### 3.4. Integration Site Analysis for P1 and P2 Samples

The transgene integration site (IS) was studied in the BM and spleen samples derived from the mice injected with P1 and P2 corrected cells.

For the mice injected with P1 cells, among the 10 most prominent IS, eight of them were overlapping between BM and spleen (Figure 7a). Three dominant IS near *SPATS2, AKAP8L* and *B4GALNT4* genes had retrieval frequencies of 31.6%/30.1%, 24%/26% and 14.1%/20.5% for the spleen and BM sample, respectively. Due to the semi-random nontargeted integration pattern of lentiviral vectors, it is unlikely that integration occurs in the exact same IS position independently in two different cells. Rather, the same IS is a marker for cells stemming from a common transduced progenitor cell. Therefore, these data support the idea that transduced human bone marrow cells migrated to the spleen. Common integration site (CIS) analysis, which clusters IS within a maximum distance of 50 kb between individual IS, revealed that 18 CIS containing 2–4 individual ISs were detected in the RAG1-transduced cells from P1 (Appendix A) suggesting a diverse composition of clusters, in line with the IS integration profile. Cancer gene analysis revealed IS within 100 kb to TSS of CG in all samples. A unique retrieval frequency of 24% (spleen) and 26% (BM) for an IS near the *BRD4* gene was observed (Appendix A). All the other ISs were below 7%. We did not find any integration of nearby genes involved in severe adverse events found in previous clinical gene therapy trials (Appendix A).

Three different NSG mice received transduced cells from P2 samples. In each of them, 3 of the 10 most prominent IS were overlapping between BM and spleen (Figure 7b). Of note, the overlapping genes are specific in each mouse. In the BM of T4 and T7 mice, *WDR60* and *PTP4A2* showed retrieval frequencies equal to 42.7% and 42.3%, respectively, but the retrieval frequencies decreased in the spleen (3% for *WDR60* and 11.8% for *PTP4A2*). Of note, the high frequency of these clones was not associated with a clonal expansion as the chimerism in P2-T4 and P2-T7 remained lower than 25% in the BM and below 3% in the spleen. CIS analysis revealed that 8.2% of IS were found in 11 CIS with a maximum of 3 IS per CIS (Appendix A). Clustered integration in or nearby genes previously involved in severe adverse events in previous gene therapy trials was not observed (Appendix A). Cancer gene analysis of the IS revealed that the IS with the highest retrieval frequency (27% on the spleen of the T4 mouse) was located near *BIRC6*. The IS analysis in transduced CD34+ HSPCs from P2 demonstrated a high polyclonality with low relative frequencies of the 10 most prominent ISs (<0.8%), 138 CISs with orders ranging from 2 to 6 (Appendix A) and no ISs were found in proximity to cancer genes (Appendix A).

Altogether, these data illustrate the highly diverse IS in all the samples without any occurrence of clonal expansion in vivo.

### 3.5. Optimization of RAG1 Gene Transfer Strategy

In all the above experiments described, the main hurdle was linked to the few numbers of bone marrow CD34+ obtained from pediatric RAG1-deficient patients and to the low cell recovery after the transduction step. This last observation was directly correlated with the huge decrease proportion of the pre-B cell subset (CD34+CD19+) after culture (Figure 1). We made the assumption that *RAG1* gene transfer preferentially targets the more immature CD34+CD19- subset. To validate this hypothesis, CD34+CD19+ and CD34+CD19- bone marrow cells from P3 patients were sorted, transduced (T) or not (NT) with the RAG1 lentiviral pre-GMP vector. As observed in Figure 8a, the number of cells in the CD34+CD19- subset increased over time whereas in the CD34+CD19+ subset, the cell number decreased. This observation is independent of the NT or T conditions. The CD34+CD19+ cells put on the T cell differentiation systems died in a few days and conversely, the RAG1-transduced CD34+CD19- cells differentiated toward the early (CD7+) T cell progenitor stage (Figure 8b). In this population, *RAG1* transgene expression was detected (Figure 8c) with a VCN close to the one obtained in the cytokine culture condition (Figure 8c), an observation highly reproducible in two other independent experiments (Figure 8d). After a 28-day culture on the OP9-DL1 system, CD4+CD8+ T cells were found with a higher frequency in the RAG1-transduced condition as compared to the NT counterpart (Figure 8e). The high mortality rate of the CD34+CD19+ population and the capacity of the RAG1-transduced CD34+CD19- subset to differentiate along the T cell lineage were confirmed on another sorted CD34+CD19- population (P5-p.Gly957Val mutation, Figure 8e). For the P3 bone marrow sample, the CD7+ cells were injected in vivo in 3-day-old NSG mice. Six weeks later, mice were killed and the thymus collected. Human chimerism was detected in all the mice and thymic reconstitution was effective in the three mice engrafted with the RAG1 corrected cells (Figure 8f). These experiments confirmed that the pro-B cell subset over-represented in RAG1 bone marrow cells cannot survive ex vivo and may explain the high mortality rate observed during the transduction step. Conversely, the CD34+CD19- subset can be efficiently transduced and differentiated both in vitro and in vivo along the T cell lineage pathway. Altogether, our data demonstrate that *RAG1* gene transfer is effective in uncommitted CD34+ and confirm the efficacy of the pre-GMP RAG1 lentiviral batch to correct RAG1 deficiency.

## 4. Discussion

Several reports have demonstrated that autologous CD34+ gene therapy can provide a good immune reconstitution for the treatment of SCID without any matched sibling donor. Although side adverse effects due to insertional mutagenesis have been reported with γ-retroviral vectors, the development of lentiviral vectors greatly improved CD34+ transduction leading to a stable immune reconstitution with a 5-year survival close to 98% [33].

In this report, we tested the safety and efficacy of a RAG1 pre-GMP lentiviral batch on bone marrow CD34+ isolated from HD or five RAG1 patients with different mutations. On HD samples, we observed that RAG1 overexpression was not toxic for CD34+ proliferation and differentiation both in vitro and in the NSG xenotransplantation model. At constant MOI, the vector concentration (Ig/mL) in the well has no impact on the VCN or on cell fate. This observation was confirmed in the two experiments with P1 cells and also described in previous reports [34,35]. In RAG1 patient’s CD34+, the mean VCN was 2.9 (1.8 to 5) after the in vitro culture (cytokine cocktail or T cell differentiation) emphasizing that *RAG1* transgene integration is stable in vitro. In the bone marrow of the NSG mice, the VCN is often lower (VCN < 2) but very comparable in the different mice that received the same cells, attesting that the transgene integration is stable in vivo. The discrepancy between the VCN number in vitro and in vivo could be explained by the heterogeneity of the HSPCs compartment, an observation already highlighted in previous reports [34,35]. Alternatively, we compared the VCN on HSPCs in two different settings: after a 14-day culture and in the total bone marrow of NSG mice, 16 weeks after the HSPCs injection. In terms of safety, the VCN observed in all the samples fulfills the criteria required for clinical use and drives effective *RAG1* expression. Moreover, the integration site analysis did not detect any integration nearby proto-oncogenes involved in severe adverse events in previous clinical gene therapy trials like *LMO2* or *MECOM*. These results are in line with a typical lentiviral profile, showing a low tendency to integrate at recurrent sites and characterized by a low enrichment of CIS [36,37,38]. Nevertheless, the reports on insertional mutagenesis in the X-ALD trail employing a lentiviral vector with a similar MND strong promoter is reason for caution. On the other hand, weaker promotors were insufficient to drive RAG1 expression in developing thymocytes to sufficient levels to regenerate αβ T cells [20].

Finally, in the RAG1-transduced conditions, the increased clonality and CDR3 length diversity associated with the presence of human serum IgM underline the presence of functional T and B lymphocyte compartments. Although the experiments were carried out on a small number of samples, these data confirmed the safety of the RAG1 pre-GMP batch and the efficacy of this RAG1 gene therapy approach to correct RAG1 deficiency.

The variability observed in the experiments performed with P1 or P2 patient’s cells was puzzling but could be attributed to the nature of the RAG1 mutations: a stop codon in the N-term part of RAG1 protein for P1 versus a compound heterozygote mutation in the C-term part for P2. Indeed, it has been reported that VDJ recombinase activity is specifically altered with a mutation affecting the N-term part (1-215 amino-acids) of the protein [39,40] suggesting that in P2 cells, a competition could occur between the RAG1 transgene and endogenous RAG1 protein in the control of the VDJ recombination step. Conversely, no VDJ recombination was detected in cells with the stop codon mutation p.Glu174Serfs × 27 identical to the one found in P1 cells [41]. Moreover, several studies shed light on RAG1 patients bearing the same mutation can have different clinical manifestations and outcomes [6,42] and RAG1 mutations driving partial RAG1 deficiency are associated with immune dysregulation [43,44] that could affect GT efficacy. Finally, the donors are also a source of variability as already observed in different GT clinical trials for PID [45,46,47]. The diversity observed among NSG mice transplanted with identical pools of transduced or non-transduced CD34+ cells highlights a notable limitation of the NSG model. It appears that only a select few genuine stem cells successfully engraft and contribute to lymphopoiesis, a process that varies significantly between individual mice. While the NSG model remains the optimal choice for preclinical testing of patient-derived cells, this inherent variability poses a significant challenge, particularly when dealing with limited quantities of patient cells.

In RAG1 patients, it is well known that the B cell differentiation arrest at the pre-B stage leads to the accumulation of a high proportion of CD34+CD19+ cells [3]. This observation prompted us to evaluate the transduction efficiency in the sorted CD34+CD19- and CD34+CD19+ subsets. In three independent experiments, we observed that in vitro proliferation and T cell differentiation were restricted to the RAG1-transduced CD34+CD19- compartment with a mean VCN comparable to the one observed in the experiments with unsorted HSPCs. Finally, the capacity of the RAG1-transduced CD34+CD19- compartment to restore the T cell compartment was confirmed in vivo in neonate NSG mice. All these data are in accordance with the previous reports observed in the RAG1-deficient mice [20] and confirm the efficacy of this pre-GMP batch to express RAG1 in bone marrow cells and to restore T and B cell differentiation in RAG1-deficient HSPCs.

In conclusion, in these preclinical studies, we demonstrated that *RAG1* gene transfer into human RAG1-deficient HSPCs restored a functional B and T lymphoid compartment in vitro and in vivo. Importantly, the efficacy of the pre-GMP batch was confirmed in several patient samples with different mutations. These data were essential to initiate the clinical GMP batch production. This innovative therapy is now approved by the European Medicines Agency (EMA) and a phase I/II clinical trial is now open for RAG1 patients with HLA-incompatible donors in the EU, UK, Turkey and Australia.

## Figures and Tables

**Figure 1 biomedicines-12-01495-f001:**
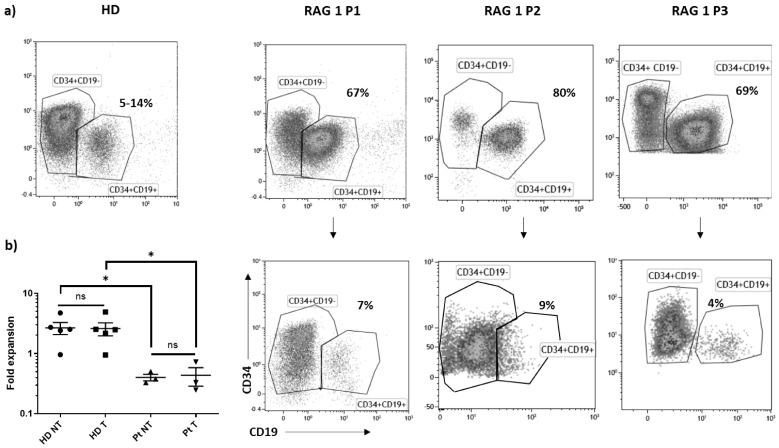
RAG1 bone-marrow samples phenotype. Healthy donor (HD) or RAG1 patient (P1, P2, P3) bone marrow CD34+ cells were transduced (T) or not (NT) with the RAG1-pre GMP lentiviral batch. (**a**) Phenotype analysis before and after the transduction step by flow cytometry. (**b**) Fold expansion in HD and patient (Pt) measured as cell number after the transduction step (Day 3)/cell number at the beginning of the culture (Day 0). * *p* < 0.05; ns: not significant.

**Figure 2 biomedicines-12-01495-f002:**
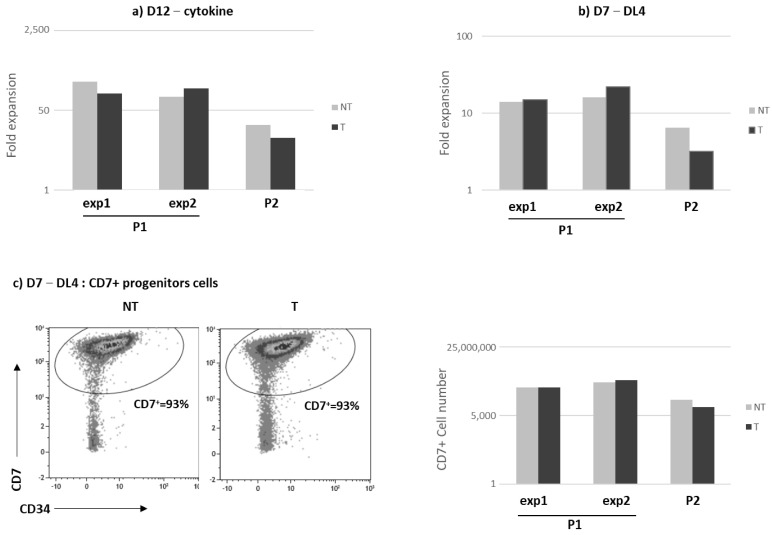
In vitro proliferation and differentiation of RAG1 gene therapy corrected HSPCs from P1 and P2 bone marrow samples. RAG1 patient (P1, P2) bone marrow CD34+ cells were transduced (T, black histogram) or not (NT, grey histogram) with the RAG1-pre GMP lentiviral batch. (**a**) Cell expansion was evaluated after a 12-day culture with cytokines in two different experiments (exp1 and exp2) for P1 patients. (**b**) Fold change cell expansion after a 7-day culture on DL4. (**c**) CD7+ percentage and cell number after a 7-day culture on DL4. (**d**) CD4+CD8+ percentage and cell number after culture on OP9-DL1 stroma for P1. (**e**) CD4+CD8+ percentage after culture on OP9-DL1 stroma for P2. The percentage of CD7, CD4 and CD8 positive cells was evaluated by flow cytometry.

**Figure 3 biomedicines-12-01495-f003:**
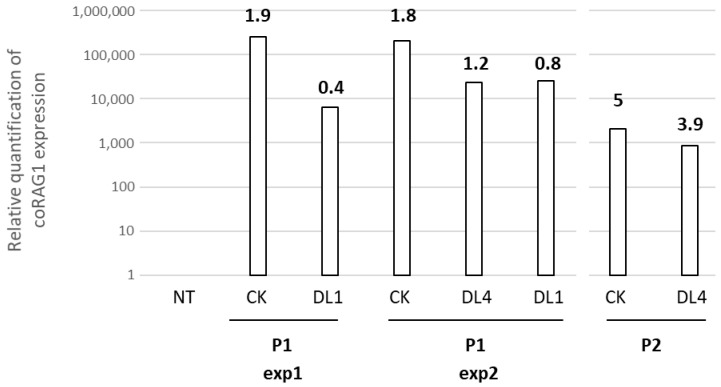
VCN and RAG1 transgene expression after in vitro cultures. Relative quantification of coRAG1 expression and VCN (number in bold) were analyzed in two different BM samples from P1 and P2 RAG1 patients after culture in the cytokine cocktail (CK), on DL4 or OP9-DL1 (DL1).

**Figure 4 biomedicines-12-01495-f004:**
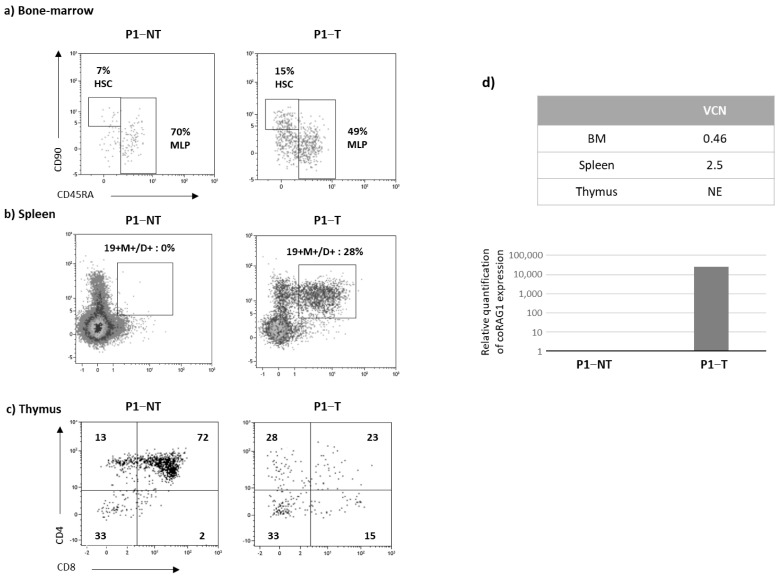
Immune reconstitution in P1 NSG mice (16 weeks). CD34+ cells from P1 patients were transduced (T) or not (NT) and injected into NSG mice. Four months after transplantation, phenotype analysis of the bone marrow (**a**), spleen (**b**) and thymus (**c**) was evaluated by flow cytometry after gating on the human CD45+ cells. (**d**) VCN evaluation in hematopoietic organs and relative quantification of coRAG1 expression in the spleen. (**e**) Diversity of the TCRB and TCRA repertoire of spleen samples from mice transplanted with P1cells. Plots show the V- and J-gene recombination of clonotypes detected in each sample. Upper plots correspond to replicates of mice transplanted with non-transduced (NT) cells and lower plots correspond to replicates of spleen samples transplanted with RAG1-transduced (T) cells. Each bow between the V- and J-gene represents a clonotype. The broader the bow, the higher the relative frequency detected for the respective clonotype. Each sample was analyzed in two replicates (replicate 1 in Figure 4e and replicate 2 in Appendix A). (**f**) CD3R length distribution in the spleen. Plots show the length of CDR3 sequences of clonotypes detected in each sample. Upper plots correspond to replicates of mice transplanted with non-transduced (NT) cells and lower plots correspond to replicates of spleen samples transplanted with RAG1-transduced (T) cells. The x-axis shows the CDR3 length in base pairs and the y-axis shows the relative frequency of the corresponding clonotype.

**Figure 5 biomedicines-12-01495-f005:**
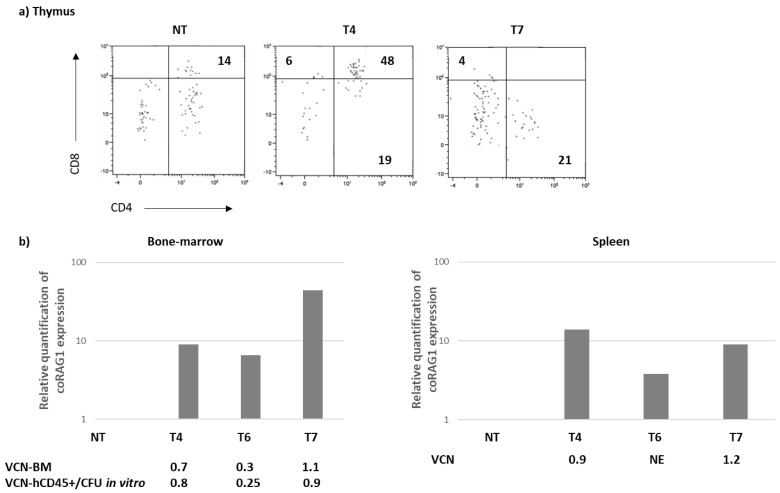
Immune reconstitution in NSG mice (16 weeks)-P2. CD34+ cells from P2 patients were transduced (T) or not (NT) and injected into NSG mice. (**a**) Four months after transplantation, the chimerism (%) and phenotype analysis were performed by flow cytometry after gating in the human CD45+ cells. (**b**) Relative quantification of coRAG1 expression and VCN evaluation in the bone marrow and spleen of the 4 transplanted mice (NT, T4, T6, T7).

**Figure 6 biomedicines-12-01495-f006:**
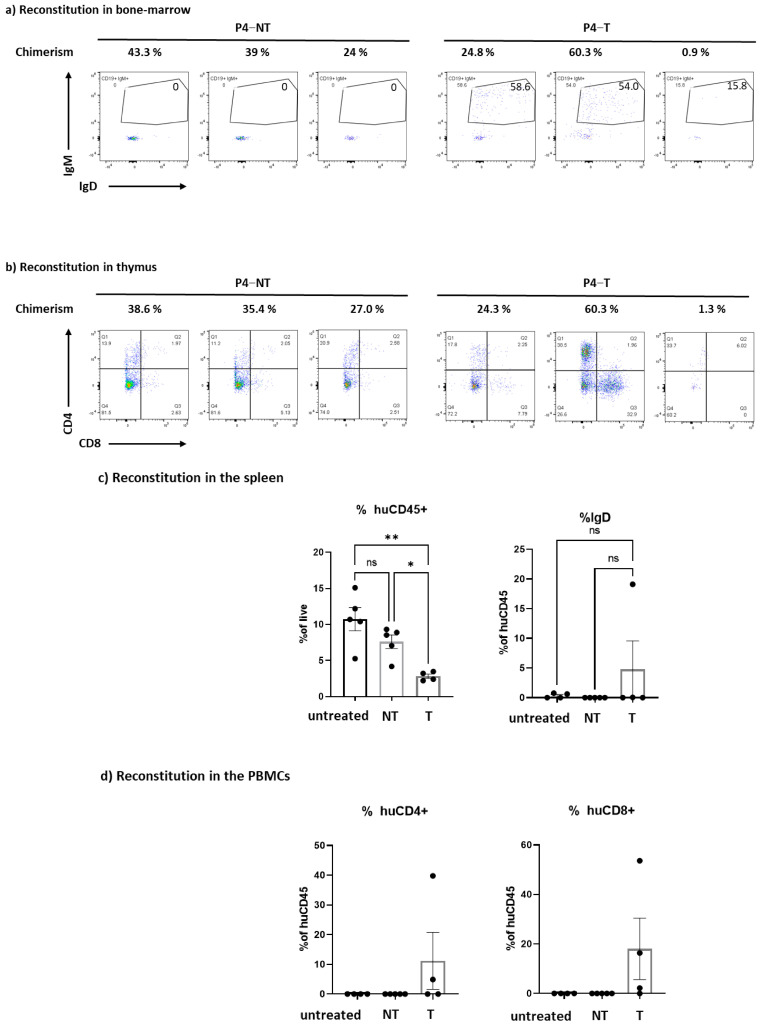
Immune reconstitution in NSG mice transplanted with HSPCs from the 1st patient included in the GT clinical trial (P4). Mobilized CD34+ cells from P4 patients were transduced (T) or not (NT) and injected into NSG mice (n = 5 mice/group). The chimerism and the phenotype were evaluated 4 months after transplantation. (**a**) Chimerism (% gated on alive cells) and human B cell reconstitution (gated on the CD19+ population) in the bone marrow. The circle represent the percentage of CD19+IgM+ cells (**b**) Chimerism (% gated on alive cells) and human T cell reconstitution in the thymus. c) Evaluation of the percentage of human CD45+ cells (% huCD45+) and immature B cell (% IgD in the human CD45+ gate) in the spleen. As an internal control, untreated (no culture, no transduction) mobilized CD34+ were also injected in the mice. (**c**) Evaluation of the percentage of human CD45+ cells (% huCD45+) and mature B cell (% IgD in the human CD45+ gate) in the spleen. As an internal control, untreated (no culture, no transduction) mobilized CD34+ were also injected in the mice. (**d**) Evaluation of the percentage of human CD4+ cells (% huCD4+) and human CD8+ cells (% huCD8+) in the peripheral blood mononuclear cells (PBMCs) after a gating on the huCD45+ population. As an internal control, untreated (no culture, no transduction) mobilized CD34+ were also injected in the mice. ns = not significant, * *p* < 0.05 ** *p* < 0.001; ns: not significant.

**Figure 7 biomedicines-12-01495-f007:**
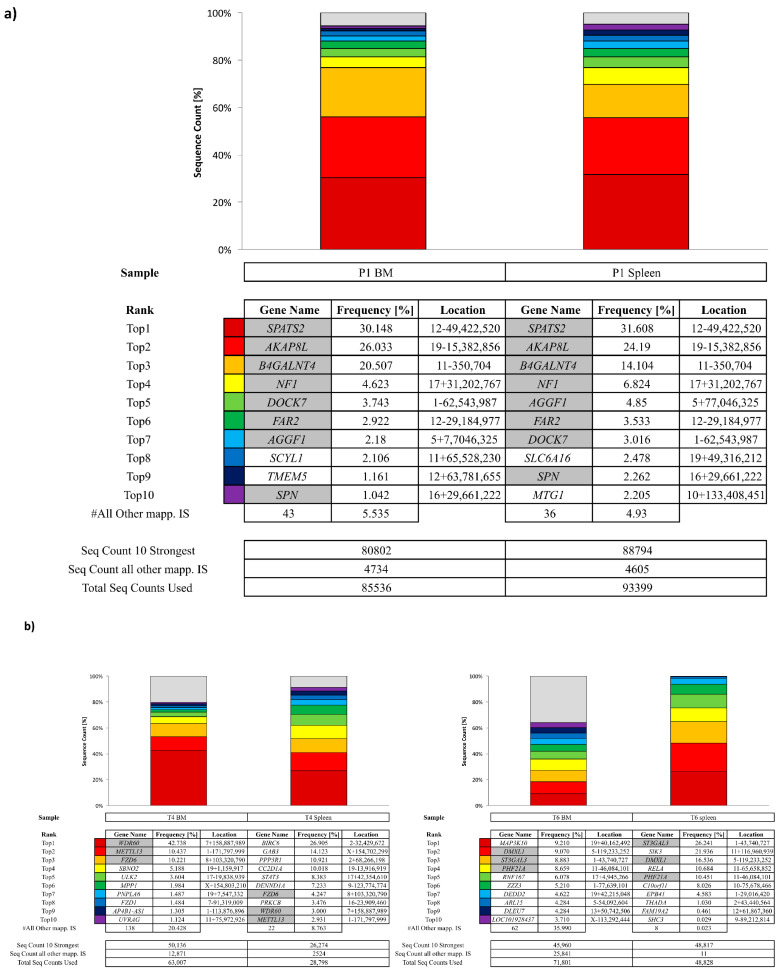
Integration site analysis in NSG mice (16 weeks). Integration site (IS) analysis was analyzed in samples from NSG mice transplanted with P1 (**a**) or P2 (**b**) and (**c**) IS analysis in CD34+ from P2 patients cells after a 14-day in vitro culture. Cumulative retrieval frequencies of the ten most prominent IS detected are shown. For individual samples, sequence data from all S-EPTS/LM-PCR replicates are combined. Sequence count of the ten most prominent IS (Seq Count 10 Strongest), sequence count of all remaining IS (Seq Count all other mapp. IS) and total IS-specific sequence count from all replicates (Total Seq Count Used) are shown at the bottom for each sample. RefSeq names of genes located closest to the respective IS are given in the (Gene Name). Relative sequence count contributions of the ten most prominent IS and all remaining mappable IS are shown (Freq., Frequency [%]). The location column is composed of chromosome number, sequence orientation (plus or minus) and IS locus (based on human reference genome hg38). Grey-marked fields represent the top 10 ISs detected in both samples.

**Figure 8 biomedicines-12-01495-f008:**
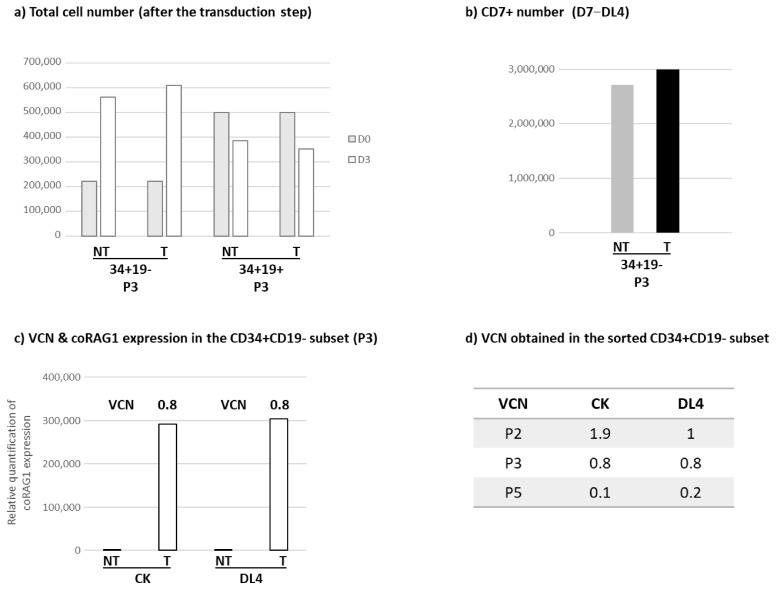
Proliferation and differentiation of the sorted CD34+CD19- and CD34+CD19+ HSPCs subsets isolated from RAG1 mutated patients. CD34+ cells from patient bone marrow samples were sorted into two groups depending on the expression of the CD19, resulting in the following subsets: CD34+CD19- and CD34+CD19+. Sorted cells were transduced (T) or not (NT) with the RAG1-pre GMP lentiviral batch. (**a**) Total cell number evaluation in CD34+CD19- and CD34+CD19+ subsets before the transduction step (D0, grey histogram) and after the transduction step (D3, white histogram). (**b**) CD7+ cell count obtained after a 7-day culture on DL4 in the NT (grey histogram) and T (black histogram) conditions. (**c**) Relative quantification of coRAG1 expression and VCN evaluation within the CD34+CD19- subset after a 14-day culture in cytokine cocktail (CK) and a 7-day culture on DL4. (**d**) VCN obtained in the transduced CD34+CD19- subset after culture in the CK or DL4 conditions. Samples from 3 patients were represented (P2, P3 and P5). (**e**) CD4+CD8+ percentage and cell number evaluation after a 28-day culture on the OP9-DL1 stroma. (**f**) Chimerism and CD4+CD8+ percentage were evaluated by flow cytometry in the thymus of six-week-old NSG mice injected with CD34+CD19-cells 3 days after birth.

**Table 1 biomedicines-12-01495-t001:** RAG1 mutations identified in the SCID patients (P1 to P5).

	RAG1 Protein
P1	p.Glu174Serfs × 27
P2	p.Thr708Ala/p.Glu669Lys
P3	p.Arg474His/p.Arg559Ser
P4	p.Glu174Serfs × 27
P5	p.Gly957Val

**Table 2 biomedicines-12-01495-t002:** Human chimerism and B cell number in NSG mice.

	Cells Injected/ Mouse	Bone-Marrow Chimerism (%)	Bone-Marrow CD19+IgM+ Number	Spleen Chimerism (%)	Spleen CD19+IgM+D+ Number	Serum Igs (ng/mL)	Thymus Chimerism (%)
P1-NT	85,000	86	68,000	38	0	24	75
P1-T	185,000	75	173,600	13	73,758	690	4
P2-NT	260,000	10.3	24,119	0.6	154	69	51.6
P2-T4	693,000	9.5	3736	2.8	2480	190	29.5
P2-T6	693,000	11.5	19,916	2.2	538	81	NE
P2-T7	693,000	23	57,553	2.4	434	85	25

NE: Non evaluable; The chimerism is calculated as follows: % human CD45^+^cells/(% mouse CD45^+^ + % human CD45^+^ cells).

**Table 3 biomedicines-12-01495-t003:** Top 10 TCRA and TCRB clones in the spleen of mice engrafted with non-transduced (NT) and RAG1-transduced cells (T) from the P1 sample.

Sample	Rank	Clonotype Frequency [%]	Recombination			Sample	Rank	Clonotype Frequency [%]	Recombination		
NT spleen replicate 1	Top 1	32.00	TRBV29-1	N/A	TRBJ2-4	T spleen replicate 1	Top 1	4.98	TRBV21-1	TRBD1	TRBJ1-2
	Top 2	20.00	TRBV27	N/A	TRBJ2-1		Top 2	2.99	TRBV6-5	TRBD1	TRBJ2-2
	Top 3	12.00	TRBV5-1	TRBD1	TRBJ1-6		Top 3	2.99	TRBV21-1	N/A	TRBJ2-7
	Top 4	8.00	TRBV7-2	TRBD1	TRBJ1-5		Top 4	2.99	TRBV29-1	TRBD2	TRBJ1-1
	Top 5	4.00	TRBV5-1	TRBD2	TRBJ2-3		Top 5	2.66	TRBV27	TRBD2	TRBJ2-2
	Top 6	4.00	TRBV5-1	TRBD1	TRBJ2-3		Top 6	2.33	TRBV29-1	N/A	TRBJ2-7
	Top 7	4.00	TRBV5-1	N/A	TRBJ2-7		Top 7	2.33	TRBV18	N/A	TRBJ2-1
	Top 8	4.00	TRBV29-1	N/A	TRBJ2-4		Top 8	1.99	TRBV29-1	N/A	TRBJ2-3
	Top 9	4.00	TRBV29-1	TRBD1	TRBJ2-7		Top 9	1.66	TRBV2	TRBD1	TRBJ2-1
	Top 10	4.00	TRBV29-1	TRBD1	TRBJ1-6		Top 10	1.66	TRBV29-1	N/A	TRBJ2-5
Sample	Rank	Clonotype Frequency [%]	Recombination			Sample	Rank	Clonotype Frequency [%]	Recombination		
NT spleen replicate 2	Top 1	14.58	TRBV20-1	TRBD2	TRBJ2-1	T replicate 2	Top 1	6.61	TRBV5-1	TRBD2	TRBJ1-1
	Top 2	14.58	TRBV29-1	TRBD1	TRBJ1-2		Top 2	4.31	TRBV7-9	N/A	TRBJ2-3
	Top 3	10.42	TRBV20-1	TRBD1	TRBJ1-2		Top 3	4.31	TRBV7-2	N/A	TRBJ2-3
	Top 4	10.42	TRBV3-1	N/A	TRBJ1-1		Top 4	4.02	TRBV7-9	N/A	TRBJ2-7
	Top 5	6.25	TRBV12-1	TRBD2	TRBJ2-4		Top 5	3.45	TRBV11-2	TRBD2	TRBJ2-7
	Top 6	6.25	TRBV18	TRBD1	TRBJ1-5		Top 6	3.45	TRBV11-2	TRBD1	TRBJ2-2
	Top 7	6.25	TRBV2	TRBD1	TRBJ2-7		Top 7	3.45	TRBV7-9	TRBD1	TRBJ2-7
	Top 8	2.08	TRBV5-1	TRBD2	TRBJ2-7		Top 8	3.16	TRBV5-1	TRBD1	TRBJ2-6
	Top 9	2.08	TRBV10-3	TRBD2	TRBJ2-5		Top 9	3.16	TRBV6-2	TRBD2	TRBJ2-1
	Top 10	2.08	TRBV5-1	TRBD2	TRBJ2-1		Top 10	3.16	TRBV6-2	TRBD1	TRBJ2-7
Sample	Rank	Clonotype Frequency [%]	Recombination			Sample	Rank	Clonotype Frequency [%]	Recombination		
NT spleen replicate 1	Top 1	38.46	TRAV12-1	N/A	TRAJ35	T spleen replicate 1	Top 1	6.84	TRAV21	N/A	TRAJ27
	Top 2	23.08	TRAV13-1	N/A	TRAJ5		Top 2	5.13	TRAV2	N/A	TRAJ31
	Top 3	23.08	TRAV13-2	N/A	TRAJ15		Top 3	4.27	TRAV12-1	N/A	TRAJ23
	Top 4	7.69	TRAV29DV5	N/A	TRAJ45		Top 4	4.27	TRAV13-1	N/A	TRAJ23
	Top 5	7.69	TRAV13-1	N/A	TRAJ41		Top 5	4.27	TRAV12-1	N/A	TRAJ27
	Top 6	N/A	N/A	N/A	N/A		Top 6	3.42	TRAV38-2DV8	TRDD3	TRAJ32
	Top 7	N/A	N/A	N/A	N/A		Top 7	2.56	TRAV13-1	N/A	TRAJ9
	Top 8	N/A	N/A	N/A	N/A		Top 8	2.56	TRAV13-1	N/A	TRAJ6
	Top 9	N/A	N/A	N/A	N/A		Top 9	2.56	TRAV12-3	N/A	TRAJ16
	Top 10	N/A	N/A	N/A	N/A		Top 10	2.56	TRAV8-4	N/A	TRAJ23
Sample	Rank	Clonotype Frequency [%]	Recombination			Sample	Rank	Clonotype Frequency [%]	Recombination		
NT spleen replicate 2	Top 1	19.4	TRAV14DV4	N/A	TRAJ13	T spleen replicate 2	Top 1	4.64	TRAV21	N/A	TRAJ12
	Top 2	12.9	TRAV13-1	N/A	TRAJ4		Top 2	3.31	TRAV21	N/A	TRAJ11
	Top 3	9.7	TRAV41	N/A	TRAJ29		Top 3	3.31	TRAV13-1	N/A	TRAJ32
	Top 4	6.5	TRAV9-2	N/A	TRAJ9		Top 4	2.65	TRAV4	N/A	TRAJ31
	Top 5	6.5	TRAV20	N/A	TRAJ57		Top 5	2.65	TRAV21	N/A	TRAJ37
	Top 6	6.5	TRAV21	N/A	TRAJ36		Top 6	2.32	TRAV21	N/A	TRAJ10
	Top 7	3.2	TRAV38-1	N/A	TRAJ58		Top 7	2.32	TRAV21	N/A	TRAJ17
	Top 8	3.2	TRAV38-2DV8	N/A	TRAJ31		Top 8	2.32	TRAV20	N/A	TRAJ8
	Top 9	3.2	TRAV20	N/A	TRAJ38		Top 9	2.32	TRAV21	N/A	TRAJ33
	Top 10	3.2	TRAV1-1	TRDD2	TRAJ26		Top 10	1.99	TRAV13-1	TRDD3	TRAJ44

## Data Availability

Data is contained within the article and Appendix A.

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
