# Peer review of "Restoration of T and B Cell Differentiation after RAG1 Gene Transfer in Human RAG1 Defective Hematopoietic Stem Cells"

_biomedicines, 2024, doi:10.3390/biomedicines12071495_

Round 1

Reviewer 1 Report

Comments and Suggestions for Authors

The manuscript by Sorel et al introduced a novel finding about restoration of T and B cell differentiation after RAG1 gene transfer in human RAG1 defective hematopoietic stem cells. Although the design is interesting, I have several concerns on the manuscript:

1.     The NSG mice is immune-defective due to knockdown of the Prkdc and Il2rg genes, is it possible to restore its immunity simply by one gene transfer? Could the authors provide more phenotypes to support the observation, or fingering out the underlying mechanism?

2.     Error bar is lost in almost all the histograms of the figures. Biological repeat is required for most of the analysis. Besides, the quality of Figure 4 could be further improved, especially the red rectangle.

3.     The shortcoming of the study is completely lost.

Comments on the Quality of English Language

no comment.

Author Response

Reviewer 1

  1. The NSG mice is immune-defective due to a knockdown of the Prkdc and Il2rg genes, it is possible to restore its immunity simply by one gene transfer? Could the authors provide more phenotypes to support the observation, or fingering out the underlying mechanism?

As mentioned in several papers, following engraftment of human HSPCs, humanized immunodeficient NSG mice are able to sustain hematopoiesis, especially for the development of the T and B lymphoid cells (Shultz LD. et al. Nat Rev Immunol, 2007; Manz MG. and Di Santo JP. Nat Immunol. 2009 ; Ishikawa F. et al. Blood, 2005, Wiekmeijer, Biores 2014).  The NSG model has been widely used for several primary immune deficiencies in order to demonstrate that gene transfer strategy is a safe and efficient approach to correct defective HSPCs (Weber et al, Mol Ther Methods Clin Dev. 2018 ; Carriglio et al, Hum Gene Ther Clin Dev, 2017, Wie, Bio Res, 2014kmeijer et al J Allergy Clin Immunol, 2016….). In RAG1 deficient patient, the T and B lymphoid cells are absent whereas the NK cells, red blood cells and myeloid compartment are not altered. Therefore, the goal of the gene transfer strategy is to reconstitute the T and B cell pool, as previously demonstrated in gene transfer strategy using RAG1 mice model (ref 15, 17, 19 in the manuscript). Altogether, these data justify why we used the NSG mice model to demonstrate that our RAG1 gene transfer strategy is safe and allow the reconstitute the T and B cell compartment impaired in RAG1 deficient cells. To clarify this point, we added several explanations in the introduction.

  1. Error bar is lost in almost all the histograms of the figures. Biological repeat is required for most of the analysis. Besides, the quality of Figure 4 could be further improved, especially the red rectangle.

Error bars are not present in the figure 2 and figure 3 as two different bone-marrow samples for P1 (ex1 and exp2) and only one bone-marrow sample for P2 were analyzed. Of note, the bone-marrow samples were obtained from pediatric samples (ie in the first year of life), explaining why few HSPCs were available. For ethical concern, additional bone-marrow aspirates were not allowed for these very sick children. As suggested by the reviewer, we improved the layout of  figure 4a, 4e and 4f.

  1. The shortcoming of the study is completely lost.

To improve the manuscript and clarify the taking home message, we added some explanations to the introduction and conclusion. However, we note that the use of NSG in vivo models and in vitro models for T cell development constitute the de factor state of the art models for assessing gene therapy in a preclinical setting

Reviewer 2 Report

Comments and Suggestions for Authors

In this manuscript, Sorel et al. tested a clinically approved lentiviral vector on patient-derived CD34+ cells in both In vitro and In vivo assays, leading to successful restoration of human B and T cell development. This certainly paves the way for further clinical trials for RAG1-SCID. Although it is not very surprising that lentiviral delivery of Rag1 transgene could rescue Rag1 deficiency causing RAG1-SCID, the authors previously took some efforts demonstrating Rag1 expression level is critical to this potential cure and a follow-up preclinical study is required as shown here. The experiment design is appropriate and the data are solid to support the conclusion. While I have a few comments needed to be addressed, I think this manuscript is suitable for publication in Biomedicines.

Minor comments:

1.     In line 205, the authors tested the impact of a serial of vector concentration on what? Did the author have concerns that the virus-diluting buffer would affect integration efficacy? Why the vector concentration was kept low? How did the authors measure the infectious viral genome? Relevant information should be added to the materials and methods section.

2.     In line 268, the authors stated that “the cell expansion was more limited and the cell number quantification was not possible”. Did the authors have an explanation for that? Is the cell number quantification not possible or not accurate?

3.     In the last paragraph of page 8, it is confusing what the authors were measuring, the exact integrated transgene copy number, or the Rag 1 mRNA for the confirmation of transgene integration?

4.     Figure 3. Please state clearly the unit of Y axis – RAG TG expression.

5.     I suggest the authors re-organized some of the Figures to make them more compact and readable.

Author Response

Reviewer 2

  1. In line 205, the authors tested the impact of a serial of vector concentration on what? Did the author have concerns that the virus-diluting buffer would affect integration efficacy? Why the vector concentration was kept low? How did the authors measure the infectious viral genome? Relevant information should be added to the materials and methods section.

The impact of different vector concentrations was tested in order to determine if the vector dilution in the medium culture has a greater impact on the VCN as compared to the transduction protocol in which the number of viral particules added per cell is considered (as described in reference 20 (Garcia-Perez L. et al. 2020)). The supplemental figure S1 confirmed that increasing the vector concentration had no impact on the integration efficacy (VCN), RAG1 transgene expression and the capacity of transduced cell to differentiate along the T cell lineage. The results regarding figure S1 have been modified in order to clarify this point.

We apologize for the typo mistake, the vector concentration was 9.5 108 to 40 108 Ig/ml.

The infectious viral genome was measured by PCR as described in the material and methods from the reference 20 (Garcia-Perez L and al. 2020). This reference is mentioned in our materials and methods section.

  1. In line 268, the authors stated that “the cell expansion was more limited and the cell number quantification was not possible”. Did the authors have an explanation for that? Is the cell number quantification not possible or not accurate?

For P2, a limited number of transduced CD34+ cells (40 000 cells for the non transduced and transduced conditions) was seeded on OP9-DL1 following the DL4 culture system. It is well known that OP9-DL1 culture is not associated with a high, the proliferation rate and therefore, no amplification of the cells was expected. This hypothesis was confirmed as in the culture well, few cell were observed under the microscope. Therefore, a cell count was not possible to perform during the 21-days culture and each week we only picked up few cells for the flow cytometry analysis. Of note, the low cell number observed along the culture justified why the culture was stopped at D21 instead of D28/D35 as observed with P1 samples.

  1. In the last paragraph of page 8, it is confusing what the authors were measuring, the exact integrated transgene copy number, or the Rag 1 mRNA for the confirmation of transgene integration?

We agree with the reviewer that our sentences were multi-interpretable. We measured the VCN by digital droplet PCR and the transgene expression using a RT-Q PCR. This paragraph was more expanded, providing more details, in the new version of the manuscript.

  1. Figure 3. Please state clearly the unit of Y axis – RAG TG expression.

As suggested by the reviewer, the unit of the Y axis was adjusted on figure 3 and now entitled relative quantification of coRAG1 expression. We also add further information in materials and methods to be more precise.

  1. I suggest the authors re-organized some of the Figures to make them more compact and readable.

We took note of the reviewer’s suggestion, and modified the following figures:

Figure 3: A more accurate name was added on the Y-axis

Figure 4a): As suggest by reviewer 1, the dot plot was modified in order to improve the quality of the figure.

Figure 4d) : the name of the Y-axis has been changed to “relative quantification of coRAG1 expression”

Figure 4e) : and f) only one replicate (replicate 1) was represented on the figure, the replicate 2 in now on figure supplemental 3. These modifications made it possible to resize the graphs and improve the quality of the figure.

Figure 5a) : the % of chimerism was transferred into the table 2

Figure 5b) : the Y-Axis title was modified to be in accordance with figure 4d

Figure 8c): the Y-Axis title was modified to be in accordance with figure 4d

Figure S1c): the Y-Axis title was modified to be in accordance with figure 4d

Figure S3: This figure is now added in order to have a figure 4 e) and f) more compact and readable.

Round 2

Reviewer 1 Report

Comments and Suggestions for Authors

The authors should provide a self evaluated shortcomings of the present study, to help the readers to reach a more objective understanding.

Comments on the Quality of English Language

none.

Author Response

The authors should provide a self evaluated shortcomings of the present study, to help the readers to reach a more objective understanding.

The present study summarizes the pre-clinical data performed on 5 different bone-marrow samples from 5 RAG1-SCID patients. After RAG1 gene transfer in HSPCs, the restoration of T and B cell differentiation prompted us to conclude that our gene therapy strategy could be transferred to RAG1 patient without any HLA-compatible donor. We agree that our conclusions are performed on a few samples but all the data are reproducible. Of note, three  gene therapy clinical trials have been initiated worldwide for SCID patients, all with a limited number of patients (for example, 5 patients were treated with the first gRV vector). In this context, having 5 pediatric bone-marrow samples in the pre-clinical studies to validate a gene therapy strategy was considered as a robust read-out for the European Medicines Agency, regulatory authorties in 6 European countries, Turkey and Australia, and  the scientific advisor board and the Data and Safety Monitoring Board associated with the ongoing RECOMB clinical study.